# Pd-Ce-O_x_/MWCNTs and Pt-Ce-O_x_/MWCNTs Composite Materials: Morphology, Microstructure, and Catalytic Properties

**DOI:** 10.3390/ma15217485

**Published:** 2022-10-25

**Authors:** Lidiya Kibis, Andrey Zadesenets, Ilia Garkul, Arina Korobova, Tatyana Kardash, Elena Slavinskaya, Olga Stonkus, Sergey Korenev, Olga Podyacheva, Andrei Boronin

**Affiliations:** 1Boreskov Institute of Catalysis, Pr. Lavrentieva 5, 630090 Novosibirsk, Russia; 2Nikolaev Institute of Inorganic Chemistry, Pr. Lavrentieva 3, 630090 Novosibirsk, Russia

**Keywords:** ceria, MWCNTs, nanocomposites, palladium, platinum

## Abstract

The composite nanomaterials based on noble metals, reducible oxides, and nanostructured carbon are considered to be perspective catalysts for many useful reactions. In the present work, multi-walled carbon nanotubes (MWCNTs) were used for the preparation of Pd-Ce-O_x_/MWCNTs and Pt-Ce-O_x_/MWCNTs catalysts comprising the active components (6 wt%Pd, 6 wt%Pt, 20 wt%CeO_2_) as highly dispersed nanoparticles, clusters, and single atoms. The application of X-ray diffraction (XRD) and high-resolution transmission electron microscopy (HRTEM) provided analysis of the samples’ morphology and structure at the atomic level. For Pd-Ce-O_x_/MWCNTs samples, the formation of PdO nanoparticles with an average crystallite size of ~8 nm was shown. Pt-Ce-O_x_/MWCNTs catalysts comprised single Pt^2+^ ions and PtO_x_ clusters less than 1 nm. A comparison of the catalytic properties of the samples showed higher activity of Pd-based catalysts in CO and CH_4_ oxidation reactions in a low-temperature range (T_50_ = 100 °C and T_50_ = 295 °C, respectively). However, oxidative pretreatment of the samples resulted in a remarkable enhancement of CO oxidation activity of Pt-Ce-O_x_/MWCNTs catalyst at T < 20 °C (33% of CO conversion at T = 0 °C), while no changes were detected for the Pd-Ce-O_x_/MWCNTs sample. The revealed catalytic effect was discussed in terms of the capability of the Pt-Ce-O_x_/MWCNTs system to form unique PtO_x_ clusters providing high catalytic activity in low-temperature CO oxidation.

## 1. Introduction

Composite nanomaterials are widely used in many fields of science and technology, including catalysis [1,2]. The application of the composites based on different types of materials, such as nanostructured carbon, oxides, and metals, is a perspective direction for the synthesis of catalysts possessing high activity, stability, and selectivity [3,4]. Presently, single and multi-walled carbon nanotubes are successfully used for the design of new composites for electronics, biomedicine, chemical sensors, energy storage, etc. [5,6]. A very promising direction is the synthesis of new composites based on carbon nanotubes for catalysis [7,8]. The use of carbon nanotubes (CNTs) as support for the stabilization of the active components (AC) has many advantages due to the relative inertness of the CNTs against aggressive reaction media and the capability to interact strongly with oxides and metals [9,10]. By varying the preparation technique, the CNTs with different properties can be prepared. The number of defects of the structure including point and extended defects with the presence of different functional groups stabilized on the surface can be adjusted [9,11,12,13]. These properties of the carbon nanotubes allow for stabilizing highly dispersed AC species in the resulting composite catalysts. The maximal dispersion of AC as single atom (SA) species on the surface of different carbon materials was discussed in the review [14] with a demonstration of a wide application of areas for SAs in catalysis. The high dispersion of the supported active components has paramount importance in practical catalysis when noble metals (NM) are used as AC. In oxidative catalysis, noble metals are usually used together with reducible oxides like TiO_2_, CeO_2,_ and others [15,16,17,18,19,20,21,22,23,24].

The interaction between NMs and reducible oxides generates lattice oxygen with high reactivity when the active components are in a highly dispersed state. Ceria is one of the most commonly used reducible oxides to study the effects of oxygen storage capacity and strong metal-support interaction (SMSI) with NM. The NM-CeO_2_ composites are often considered as model systems for establishing the nature of the active sites in many catalytic reactions. Ceria nanoparticles with a sufficient concentration of surface defects can strongly interact with metals. Due to SMSI, various dispersed forms of Pd and Pt such as single atoms, clusters, and sub-nanometer particles can be stabilized [25,26,27,28]. These Pd(Pt)/CeO_2_ surface species play a vital role in establishing the catalytic mechanisms. The investigation of Pd(Pt)-ceria systems depending on their composition, size, and defect structure is of high importance for catalysis. The deposition of Pd(Pt)-ceria species on the surface of carbon materials can be considered as a prospective approach that allows for obtaining detailed information on the local structure and oxidation states of the active components. The stabilization of NM on the surface of oxides in a highly dispersed state is a rather complicated task that can be solved by the deposition of NM oxide nanoparticles on the surface of suitable support, preventing bulk interaction, such as carbon nanomaterials. The specificity of NM oxides deposition is suggested to be controlled by the interaction with the surface of carbons of different defectiveness.

In our work, we studied the composite materials based on the multi-walled carbon nanotubes, CeO_2_, and NMs (Pd or Pt). The samples were prepared by impregnation of the MWCNTs with active component precursors followed by thermal treatment in an inert and oxidative atmosphere. The structural (N_2_ adsorption, XRD, HRTEM) and spectral (XPS) methods were applied for the samples’ characterization. The catalytic properties of the samples were analyzed in the temperature-programmed reaction of CO and CH_4_ oxidation (TPR-CO+O_2_ and TPR-CH_4_+O_2_, respectively).

## 2. Materials and Methods

### 2.1. Samples Preparation

#### 2.1.1. Support Preparation

The multi-walled carbon nanotubes (MWCNTs) were prepared by a standard chemical vapor deposition (CVD) method in a flow reactor with a fluidized catalyst bed using ethylene and 62wt%Fe-8wt%Ni-30wt%Al_2_O_3_ catalyst [29]. The C_2_H_4_ decomposition was performed at 700 °C. As-synthesized MWCNTs were washed in hydrochloric acid; for details, see [29].

#### 2.1.2. Pd-Based Catalysts

The 6 wt%Pd/20 wt%CeO_2_/MWCNTs samples were prepared using palladium nitrate Pd(NO_3_)_2_ 2H_2_O (prepared following the procedure described by Khranenko et al. [30]) and ammonium cerium(IV) nitrate (NH_4_)_2_[Ce(NO_3_)_6_] (CAN, Rare metals plant, Novosibirsk, Russia). The calculated amounts of the precursors of the active components were dissolved in an acetone solution and mixed with MWCNTs for 15 min at room temperature. The resulted suspension was kept for 12 h in a closed container. The acetone was removed by evaporation on air followed by drying under a vacuum. The resulting sample was heated in He atmosphere up to 200 °C with a heating rate of 10 °C/min, followed by heating up to 350 °C with a heating rate of 2 °C/min and keeping the samples at 350 °C for 30 min. The prepared sample is labeled further in the text as Pd-Ce-C. To test the influence of the oxidative pretreatment on the sample properties, a part of the Pd-Ce-C sample was further heated in 1vol%O_2_/He at 350 °C for 1 h. The prepared sample is labeled further in the text as Pd-Ce-C-ox. X-ray fluorescence spectroscopy confirmed the nominal content of the metals in the Pd-Ce-C sample. The results gave the palladium content as 6.2 ± 0.3 wt%, and ceria content as 21 ± 1 wt%. As our preparation procedure does not include any steps, where the active component can be lost such as filtration, decantation, etc., we can be sure that the content of the active components in the samples corresponds to the nominal one.

#### 2.1.3. Pt-Based Catalysts

Tetraalkylammonium salt of binuclear platinum nitrate complex ((NMe_4_)_2_[Pt_2_(µ-OH)_2_(NO_3_)_8_]) and CAN were used as precursors of the active components for the synthesis of the 6 wt%Pt/20 wt%CeO_2_/MWCNTs samples. (NMe_4_)_2_[Pt_2_(µ-OH)_2_(NO_3_)_8_] was prepared following the procedure developed by Vasilchenko et al. [31]. The calculated amounts of (NMe_4_)_2_[Pt_2_(µ-OH)_2_(NO_3_)_8_] and CAN were dissolved in acetone with further addition of the MWCNTs to the resulting solution. The suspension was processed in an ultrasound bath and left for 12 h in a tightly closed container. The solvent was removed by evaporation until visible traces of the liquid disappeared, and then by heating at 55 °C. The dried sample was heated in He atmosphere up to 200 °C (heating rate 10 °C/min) followed by heating up to 350 °C (heating rate 2 °C/min), and the samples were kept at 350 °C for 30 min. The Pt-based sample is denoted in the text as Pt-Ce-C. A part of the Pt-Ce-C sample was further heated in 1vol%O_2_/He at 350 °C for 1 h (Pt-Ce-C-ox sample).

### 2.2. Characterization by Physicochemical Methods

Specific surface area (S_BET_), pore volume, and average pore diameter were studied by N_2_ adsorption at low temperatures using ASAP 2400 automatic setup (Micrometrics, Norcross, GA, USA). Before the analysis, the samples were evacuated at 200 °C for 24 h. Analysis of the experimental data was performed in the Quantachrome Instruments program (version 2.02). The accuracy of the morphology parameters determination did not exceed 10%.

X-ray diffraction patterns were obtained on a STOE STADI MP diffractometer (Darmstadt, Germany) using MoK_α1_ radiation (λ = 0.70926 Å). A curved Ge(111) monochromator was used to generate the primary beam. The experiments were carried out in a transmission mode. The DECTRIS MYTHEN detector was used to register the signal. The diffraction patterns were collected at room temperature over 2θ range 2θ = 5–40°, step 0.0015°, and accumulation time—10 s.

The phase analysis was carried out using the ICDD PDF-2 (2009) database. Full-profile modeling by the Rietveld method for estimation of the size of coherent scattering domains (average crystallite size, *D*) and the lattice microstrain parameter (Δd/d) was performed using the Topas v4.2 program (Bruker, Karlsruhe, Germany). The instrumental broadening was taken into account using an external standard, LaB_6_, analyzed under similar conditions. As an assessment of the quality of the modeling of diffraction patterns, a weighted R factor was used.

TEM data were obtained using a double aberration-corrected Thermo Fisher Scientific (Waltham, MA, USA) Themis Z electron microscope operated at 200 kV (0.7 Å resolution). Images with a high atomic number contrast were acquired using a high-angle annular dark field (HAADF) detector in Scanning-TEM (STEM) mode. The spectrum imaging data were obtained using a Super-X G2 EDX detector (Thermo Fisher Scientific, Waltham, MA, USA). The samples for the TEM study were dispersed ultrasonically and deposited on copper grids covered with a holey carbon film.

The X-ray photoelectron spectra were collected on an ES 3000 (Kratos Analytical, Manchester, UK) photoelectron spectrometer using MgKα (hν = 1253.6 eV) X-ray source. The catalysts were fixed on the sample holder with scotch tape. The survey spectra in a range of 0–1100 eV were acquired to control the qualitative composition of the samples’ surface. The C1s, Pd3d, Pt4f, O1s, and Ce3d core-level spectra were collected to analyze the oxidation states of the elements. The C1s peak with binding energy (E_b_) 284.4 eV was used as an internal reference to check the possible charging effects. The accuracy of the E_b_ peaks position is ±0.1 eV. The experimental spectra were processed in the XPS-Calc program tested previously on the carbon-based and ceria-based catalysts [11,17]. The analysis included Shirley background subtraction followed by fitting the spectra with a combination of the Gaussian and Lorentzian functions. The accuracy of the Pd3d and Pt4f peaks fitting does not exceed 3%, and 5% in the case of the Ce3d spectra.

### 2.3. Catalytic Activity Tests

The study of the catalytic properties in the reactions of CO oxidation and CH_4_ oxidation was carried out in a setup with a flow reactor in the temperature-programmed reaction mode, the TPR-CO+O_2_ and TPR-CH_4_+O_2_ experiments, respectively. Detailed information on the catalytic setup can be found in [32]. The concentrations of CO, CO_2_, CH_4_, O_2_, H_2_, and H_2_O were measured using a quadrupole mass spectrometer RGA 200 (SRS). The catalytic properties were studied under the experimental conditions listed below. The particle size of the catalyst was 0.25–0.5 mm. The catalyst weight was 0.1 g. In the TPR-CO+O_2_ experiments, the reaction mixture rate and gas-hourly space velocity (GHSV) were 1000 mL/min and 600,000 mL/g h, respectively. The reaction mixture contained 0.2 vol.% CO, 1 vol.% O_2_, 0.5 vol.% Ne, and He (balance). Before heating in the reaction mixture, the catalysts were cooled to −40 °C. In the TPR-CH_4_+O_2_ experiments, the reaction mixture rate and GHSV were 100 mL/min and 60,000 mL/g h, respectively. The reaction mixture contained 0.1 vol.% CH_4_, 1 vol.% O_2_, 0.5 vol.% Ne, and He (balance). The samples were heated in the reaction mixture starting from 50 °C. The temperature-programmed reaction experiments included several heating cycles of the catalysts in the reaction mixture with intermediate cooling in the temperature range from −40 (+50) °C to 350 °C with a heating rate of 10 °C/min.

## 3. Results and Discussion

### 3.1. Textural Properties and XRD Data

Nitrogen adsorption–desorption isotherms for all samples belong to type IV isotherms in the IUPAC classification, typical of mesoporous materials (Figure A1). The specific surface area (S_BET_) of the Pd-Ce-C and Pt-Ce-C samples (Table 1) is comparable with the S_BET_ of the pristine MWCNTs. Oxidative pretreatment of the samples results in some increase in the S_BET_ value. The mean pore diameter in the catalysts varies in the interval 14–24 nm, and the pore volume varies in the interval 0.7–1.0 cm^3^/g.

Figure 1 presents XRD data for the analyzed samples. The major peaks on the XRD patterns of the samples can be attributed to the CeO_2_ phase (ICDD PDF-2 #00-034-0394) and carbon. Additional lines corresponding to the PdO oxide phase (ICDD PDF-2 #00-41-1107) and metallic Pt phase (ICDD PDF-2 #00-004-0802) can be detected on the patterns of the Pd-based and Pt-based catalysts, respectively.

In order to calculate the structural parameters of the detected phases, a Rietveld refinement was performed. The scattering from the MWCNTs was modeled using a set of peaks, of which position, width, and relative intensity were taken from the profile refinement of the MWCNT sample. The obtained results are given in Table 1. The Pd-Ce-C and Pt-Ce-C catalysts are characterized by small CeO_2_ particles of about 3 nm in size. The cell parameter (*a*) of CeO_2_ (*a*~5.43 Å) is higher than the one typical for the bulk CeO_2_ (*a*(CeO_2_) = 5.411 Å, ICDD PDF-2 #043-1002). The increased *a*(CeO_2_) value and the microstrain parameter Δd/d ≈ 0.3 indicate the high defectiveness of the CeO_2_ particles in the samples.

An increase in the CeO_2_ particle size and a decrease in the microstrain parameter are observed after oxidative pretreatment for the Pd-Ce-C sample. However, the size of the PdO particles (D(PdO)~8 nm) remains the same for the Pd-Ce-C and Pd-Ce-C-ox samples. After oxidative pretreatment of the Pt-Ce-C sample, no significant changes in the CeO_2_ particle size can be detected, but the intensity of the reflections of the metallic Pt phase decreases, which may be an indication of the formation of highly dispersed oxidized platinum species. The data of the refinement of the X-ray diffraction patterns by the Rietveld method was used to estimate the Pt^0^ phase content in the Pt-based samples. The Pt^0^ phase content for Pt-Ce-C and Pt-Ce-C-ox samples was about 2.6 wt% and 0.6 wt% if we consider only Pt and CeO_2_ phases. Thus, in the Pt-Ce-O_x_/MWCNTs composites, the Pt^0^ phase content does not exceed 0.5 wt.%. Therefore, the main part of platinum is not detected by XRD and is likely to be in a highly dispersed state.

### 3.2. TEM Data

According to TEM data, the Pd-Ce-C catalyst contains ceria and palladium oxide nanoparticles anchored either as single particles on the carbon surface or forming agglomerates that are in contact with MWCNTs (Figure 2a,b). PdO nanoparticles in the composition of the agglomerates have a shape close to spherical (Figure 2c), while the individual PdO nanoparticles fixed on the carbon surface have a somewhat flattened shape (Figure 2d). The size of the PdO nanoparticles varies in the range of 5–10 nm. The PdO nanoparticles are monocrystalline so their size observed by TEM corresponds well to the average crystallite size of the particles obtained by XRD (Table 1). As can be seen from the EDX mapping data and high-resolution images, the PdO particles in the agglomerates do not come into contact with each other; they are surrounded by CeO_2_ crystallites, which are somewhat smaller—from 2 to 5 nm (Figure 2a–c). The high-resolution image given in Figure 2e shows that on the surface of a single PdO particle the nanoscale CeO_2_ crystallites as well as amorphous ceria fragments are located. The presence of Ce in the composition of these fragments is confirmed by the data of high-resolution EDX mapping (Figure 2f). TEM images show that the fixation of the PdO and CeO_2_ nanoparticles on the MWCNT surface resulted in the formation of extended defects in the external graphene layers (Figure 2d). Additionally, the fixation of PdO and CeO_2_ nanoparticles can occur inside the tube channel (see Figure A2). However, the number of such nanoparticles is small.

In the Pt-Ce-C catalyst, single and agglomerated crystalline nanoparticles are observed on the surface of the MWCNTs (Figure 3). However, the size of the agglomerates in this sample is noticeably smaller and does not exceed 20 nm, whereas, in the Pd-Ce-C sample, agglomerates of about 100 nm were observed. An analysis of the interplanar distances from the high-resolution images indicates that CeO_2_ nanoparticles of about 2–5 nm and metallic platinum particles of about 1 nm are fixed on the surface of the MWCNTs (Figure 3c). At the same time, sub-nanometer Pt clusters (marked in Figure 3c,d with yellow arrows) can be seen on the surface of CeO_2_ nanoparticles. The crystal structure of these clusters is often not visible because of their small size. These clusters are best visualized in HAADF-STEM images due to the Z-contrast against the ceria background (Figure 3d). It can be seen that the platinum clusters are localized predominantly along the interblock boundaries of CeO_2_ nanoparticles. In addition, the bright spots observed on the image of the CeO_2_ crystal lattice (yellow circles in Figure 3d) suggest that some of the platinum is stabilized in the fluorite structure as single atoms or groups of atoms. Individual Pt and Ce atoms, either as single sites or in the form of small groups, are also fixed on the surface of the MWCNTs (marked by red circles in Figure 3d). Thus, the Pt-Ce-C catalyst is characterized by a dispersed state of the noble metal: nanoparticles and clusters up to 1 nm in size, single atom species. Stabilization of the platinum species occurs on the surface of both MWCNTs and ceria particles.

### 3.3. XPS Data

Figure 4 presents the Pd3d and Pt4f spectra of the corresponding Pd-based and Pt-based samples.

For both the Pd-Ce-C and Pd-Ce-C-ox samples, there is one main spin-orbit split Pd3d_5/2_-Pd3d_3/2_ doublet in Pd3d spectrum with E_b_(Pd3d_5/2_) at 337.2–337.3 eV. According to the literature data, this E_b_(Pd3d_5/2_) value is between the E_b_(Pd3d_5/2_) values typical for bulk PdO oxide (E_b_(Pd3d_5/2_) = 336.9 eV [33,34,35]) and Pd^2+^ ions incorporated in CeO_2_ matrix (E_b_(Pd3d_5/2_) = 337.8–338.2 eV [17,19,36,37]). Based on the XRD and TEM data we can interpret the observed peak as PdO nanoparticles stabilized on the surface of MWCNTs or in the composition of PdO-CeO_2_ agglomerates. The low-intensity Pd3d_5/2_-Pd3d_3/2_ doublet with E_b_(Pd3d_5/2_)~340.0 eV can be associated with the satellite structure of PdO oxide [35].

The Pt4f spectrum of the Pt-Ce-C sample shows a main Pt4f_7/2_-Pt4f_5/2_ doublet with E_b_(Pt4f_7/2_) = 73.0 eV. This doublet can be assigned to the Pt^2+^ single ions stabilized on the surface of CeO_2_ particles [27,36]. A slight shift of this doublet to lower E_b_ values observed for the Pt-Ce-C-ox sample can be caused by the association of the single Pt^2+^ ions with the formation of PtO_x_ sub-nanometer clusters. The low-intensity Pt4f_7/2_-Pt4f_5/2_ doublet with E_b_(Pt4f_7/2_) at 74.6–74.8 eV corresponds to Pt^4+^ species [27,38]. The Pt4f spectrum of the Pt-Ce-C sample shows also the low-intensity Pt4f_7/2_-Pt4f_5/2_ doublet with E_b_(Pt4f_7/2_) = 71.8 eV, which can be assigned to a highly dispersed Pt^0^ or Pt^δ+^ species [27,39,40].

The C1s spectra of the samples (Figure A3a) have a line shape typical for the sp^2^-hybrid carbon species with a maximum of the C1s peak at 284.4 eV and a broad low-intensity peak at 290–292 eV corresponding to π-π* satellite [41]. The Ce3d spectra for all analyzed samples (Figure A3b) have a structure typical for the CeO_2_-based catalysts. The spectra were fitted with individual components of Ce^3+^ (V^0^, V’, U^0^, U’ peaks) and Ce^4+^ (V, V’’, V’’’, U, U’’, U’’’ peaks) species. As can be seen from the Ce3d spectra, both Ce^3+^ and Ce^4+^ species are present in the samples. The fraction of the Ce^3+^ species was calculated as a ratio of the areas of the peaks, corresponding to the Ce^3+^ species (V^0^, V’, U^0^, U’ peaks) to the area of the overall Ce3d spectrum. The fraction of Ce^3+^ species in the samples is within the range of 15–17%.

Thus, the results of the physicochemical methods show that deposition of the active components on the surface of the MWCNTs results in the formation of the nanocomposite materials. In the case of the Pd-based samples, stabilization of the PdO and CeO_2_ nanoparticles, as well as PdO-CeO_2_ agglomerates on the MWCNTs surface takes place. For the Pt-based samples, more dispersed NM species are formed: Pt^2+^ single ions and PtO_x_ clusters less than 1 nm in size.

### 3.4. Catalytic Properties

The catalytic activity of the Pd-Ce-O_x_/MWCNTs and Pt-Ce-O_x_/MWCNTs samples treated in an inert and oxidizing atmosphere was tested in the reaction of CO oxidation. Figure 5 shows the temperature dependencies of CO conversion for the Pd-based and Pt-based catalysts. The data of the third heating run of the samples in the reaction mixture are provided. Successive heating runs of the samples in the TPR-CO+O_2_ mode make it possible to trace both the initial activity of the samples, as well as to establish the change of the catalytic characteristics during the reaction and their stability. The data collected during the third heating run corresponds to the stable state of the catalysts.

For the Pd-based catalysts, the observed temperature dependence of CO conversion shows a classical S-shaped behavior (Figure 5a). CO conversion is observed already at temperatures below 20–25 °C. The temperatures of 10 and 50% of CO conversion (T_10_ and T_50_ values) are 35 and 100 °C, respectively. It should be noted that the temperature dependences of CO conversion practically coincide during the second and third heating runs in the reaction mixture (see Figure A4), pointing to stability of the catalytic characteristics. The TPR-CO+O_2_ data for the initial sample (Pd-Ce-C) and the sample treated in 1%O_2_/He (Pd-Ce-C-ox) are very similar. Thus, pretreatment of the Pd-based sample in 1%O_2_/He does not result in any noticeable changes in the catalytic characteristics. We can conclude that the Pd-Ce-O_x_/MWCNTs nanocomposite is characterized by high stability during heating/cooling cycles in the reaction mixture up to 350 °C and its catalytic characteristics are not influenced by oxidative pretreatment.

For the Pt-Ce-C catalyst, the temperature dependence of CO also shows S-shaped behavior (Figure 5b). The temperature dependences of CO conversion practically coincide during the second and third heating runs in the reaction mixture (Figure A4). However, the obtained catalytic characteristics are somewhat worse than the ones for the Pd-based catalyst. Thus, the T_10_ and T_50_ values for the Pt-Ce-C catalyst are 70 and 110 °C, respectively. It is noteworthy that treatment of the Pt-Ce-C catalyst in 1%O_2_/He leads to significant changes in the temperature dependence of CO conversion. The U-shape behavior of the CO conversion curve is observed (see data for Pt-Ce-C-ox sample in Figure 5b). In the temperature range from −40 to 10 °C, an increase in CO conversion can be seen. However, further increase in the temperature results in the gradual decline of CO conversion until the temperature reaches 60 °C. At temperatures above 60 °C, an increase in CO conversion is observed again, and the CO conversion curve practically coincides with the one for the Pt-Ce-C sample. Thus, for the Pt-based sample, the pretreatment in 1%O_2_/He atmosphere plays a key role in the improvement of the sample activity at the low-temperature range.

The data of physicochemical methods show little changes in the structure of the Pd-based samples after oxidative pretreatment. The XRD data demonstrate the presence of the PdO nanoparticles with an average crystallite size of about 8 nm in both Pd-Ce-C and Pd-Ce-C-ox samples. The results of the XPS method also indicate the similar nature of the palladium species in both Pd-based samples. As a result, the activities of Pd-Ce-C and Pd-Ce-C-ox samples are very similar. In the case of Pt-Ce-O_x_/MWCNTs nanocomposites, much smaller oxidized Pt species are formed. TEM data indicate the formation of single Pt^2+^ ions and PtO_x_ sub-nanometer clusters less than 1 nm in size. It can be proposed that oxidative pretreatment might increase the relative number of PtO_x_ clusters. The change in the catalysts’ structure should influence the catalytic properties. The U-shape behavior of the CO conversion curve observed for the Pt-Ce-C-ox sample indicates the change in the reaction mechanism with an increase in the temperature. Such variation in the reaction mechanism is caused by the presence of several types of active sites, which are active in different temperature ranges. Based on our previous work [27], it can be assumed that at temperatures above 60 °C the Pt^2+^ single ions stabilized on the CeO_2_ surface are responsible for CO conversion, and the reaction proceeds via the Mars-van Krevelen (MvK) mechanism. At temperatures below 60 °C the sub-nanometer PtO_x_ clusters are likely to ensure the CO oxidation via the concerted Low-T mechanism [27]. So, the increase in the number of PtO_x_ clusters by oxidative pretreatment of the Pt-Ce-C sample results in the improvement of the low-temperature activity of the Pt-based samples. It should be noted that the observed behavior of the Pt-Ce-C-ox catalyst is maintained during cyclic heating in the reaction mixture up to 350 °C (Figure A3), which indicates high stability of the formed PtO_x_–based active sites.

The activity of the Pd-Ce-C-ox and Pt-Ce-C-ox catalysts was also tested in the methane oxidation reaction. The samples demonstrated high stability of the catalytic characteristics during successive heating runs (Figure A5). Figure 6 shows the TPR-CH_4_+O_2_ data collected during the third heating run of the samples in the reaction mixture.

For the Pd-Ce-C-ox and Pt-Ce-C-ox catalysts, the CH_4_ conversion starts at 200 and 250 °C, respectively. A difference in the T_50_ values of about 50 °C is also observed. These data indicate a higher activity of the Pd-Ce-C-ox catalyst compared to the Pt-Ce-C-ox catalyst. It should be mentioned that the samples able to oxidize methane at a temperature below 300 °C are considered highly active catalysts [42,43,44]. Xiong et al. [45] have recently reported the CH_4_ conversion starting from 200 °C on 1Pd/2Pt@CeO_2_ catalyst. The authors explained the high activity of their catalysts by the formation of the two-dimensional PdO_x_ rafts. It is known that PdO species are characterized by high activity in methane oxidation, and surface oxide has been shown to be more active than bulk PdO [46,47]. So, it can be concluded that the structure of the PdO species is of high importance for efficient CH_4_ conversion. In our Pd-Ce-O_x_/MWCNTs samples, the formation of the mixed Pd-Ce-O_x_ agglomerates was shown by TEM (Figure 2e,f). These agglomerates have a complex structure, so, a formation of the PdO_x_ species of a specific structure cannot be excluded. However, further experiments with a detailed analysis of the structure of these species are required to draw any reliable conclusions.

## 4. Conclusions

The MWCNTs were applied as a support for the preparation of Pd-Ce-O_x_/MWCNTs and Pt-Ce-O_x_/MWCNTs nanocomposites. The NM and cerium precursors were deposited on the MWCNTs surface from the acetone solution. Pretreatment of the samples in the inert and oxidizing atmospheres was performed. The Pd-based samples treated in both inert and oxidizing atmospheres comprise individual PdO and CeO_2_ nanoparticles less than 10 nm in size as well as PdO-CeO_2_ agglomerates. These systems show high activity in CO and CH_4_ oxidation reactions starting from 0° and 200 °C, respectively. The Pt-Ce-O_x_/MWCNTs systems contain highly dispersed platinum species mostly in a form of Pt^2+^ single ions and PtO_x_ sub-nanometer clusters. Oxidative pretreatment leads to the significant improvement of the activity of the Pt-based samples in CO oxidation reaction at a temperature below 0 °C, most likely due to the increase in the relative number of the PtO_x_ sub-nanometer clusters. So, we can conclude that the NM-Ce-O_x_/MWCNTs nanocomposites are prospective systems for various catalytic applications. Depending on the nature of the noble metal, the active component forms different stable structures. Thus, the Pd-Ce-O_x_/MWCNTs composites can be considered suitable candidates for methane oxidation due to the formation of PdO nanoparticles stabilized on the CeO_2_ and/or MWCNTs surface. The Pt-Ce-O_x_/MWCNTs species are highly active in CO oxidation at a temperature below 0 °C owing to the stabilization of the PtO_x_ sub-nanometer clusters. The further development of the current work would include studying the influence of the structure and morphology of the carbon-based support on the properties of the resulting nanocomposites. The different nanostructured carbon materials can be considered as well as MWCNTs modified by the heteroatoms.

## Figures and Tables

**Figure 1 materials-15-07485-f001:**
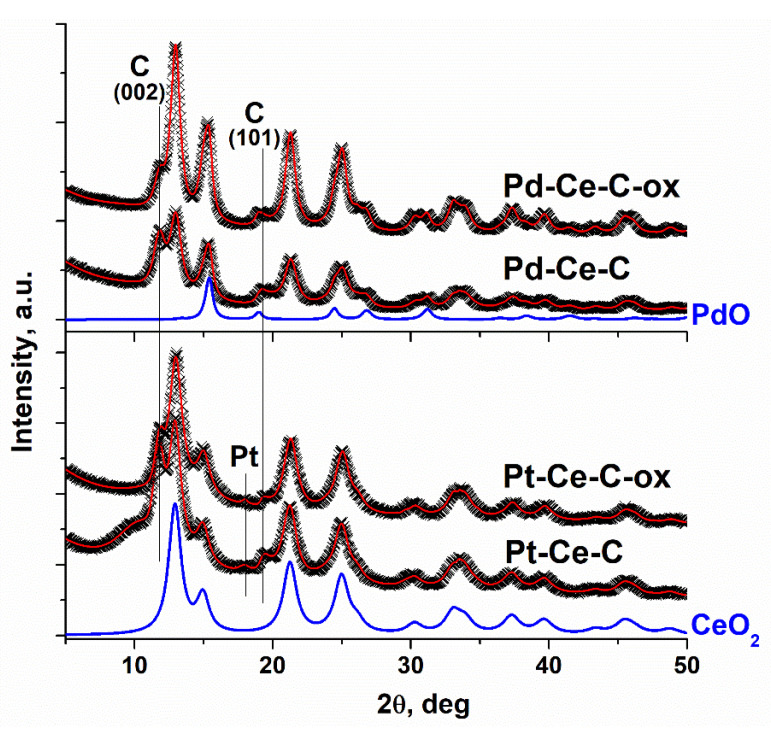
XRD data for the Pd-based and Pt-based samples. Experimental data ((x) lines), Rietveld.

**Figure 2 materials-15-07485-f002:**
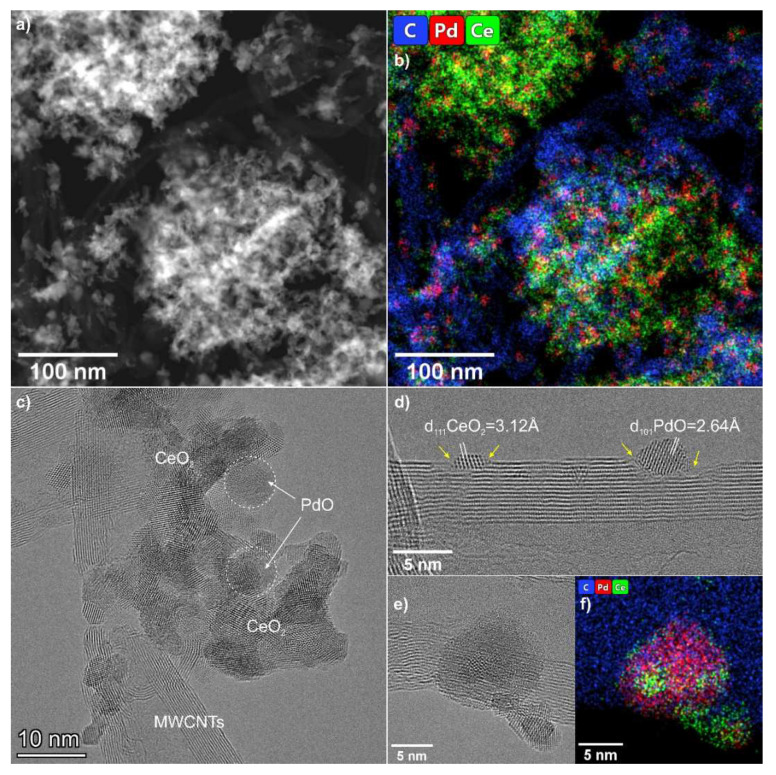
TEM data for Pd-Ce-C catalyst: (**a**,**b**) HAADF-STEM image and corresponding EDX-mapping patterns showing distribution of C (blue), Pd (red) and Ce (green); (**c**–**e**) HRTEM images; (**f**) EDX-mapping patterns for the region shown in panel (**e**). The EDX maps are presented in background-corrected intensities.

**Figure 3 materials-15-07485-f003:**
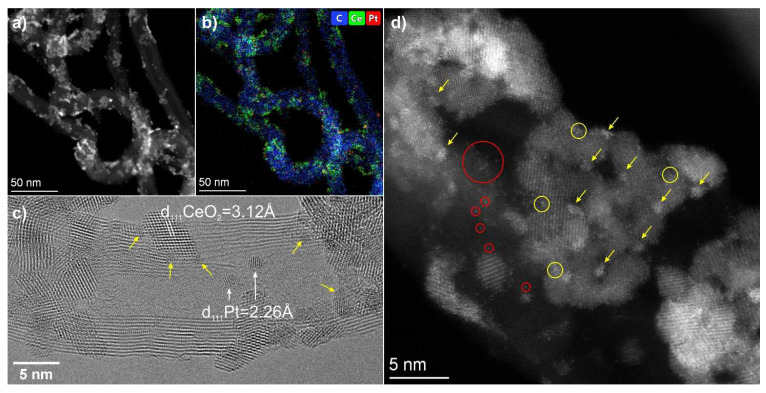
TEM data for Pt-Ce-C catalyst: (**a**,**b**) HAADF-STEM image and corresponding EDX-mapping patterns showing distribution of C (blue), Pt (red) and Ce (green); (**c**) HRTEM image; (**d**) HAADF-STEM image. The yellow arrows in images indicate the location of Pt sub-nanometer clusters on the ceria surface; the red circles mark the location of single atoms (Pt and Ce) and grouped single atoms on the MWCNT surface; the yellow circles indicate individual and grouped single Pt atoms on the ceria surface.

**Figure 4 materials-15-07485-f004:**
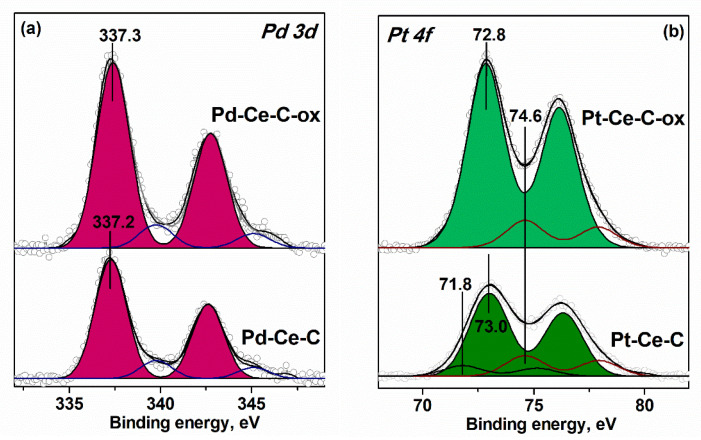
XPS data. (**a**) Pd3d spectra of Pd-Ce-C and Pd-Ce-C-ox samples; (**b**) Pt4f spectra of Pt-Ce-C and Pt-Ce-C-ox samples. ((o-o) lines correspond to the experimental spectra).

**Figure 5 materials-15-07485-f005:**
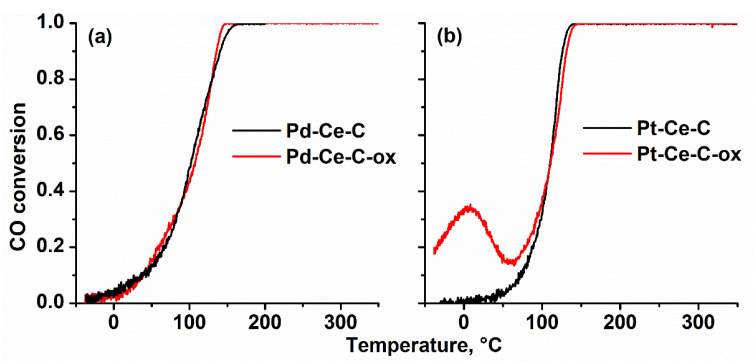
TPR-CO+O_2_ data. The temperature dependences of CO conversion during the third heating run of the samples in the reaction mixture in the TPR-CO+O_2_ mode. (**a**) Pd-Ce-C and Pd-Ce-C-ox catalysts; (**b**) Pt-Ce-C and Pt-Ce-C-ox catalysts.

**Figure 6 materials-15-07485-f006:**
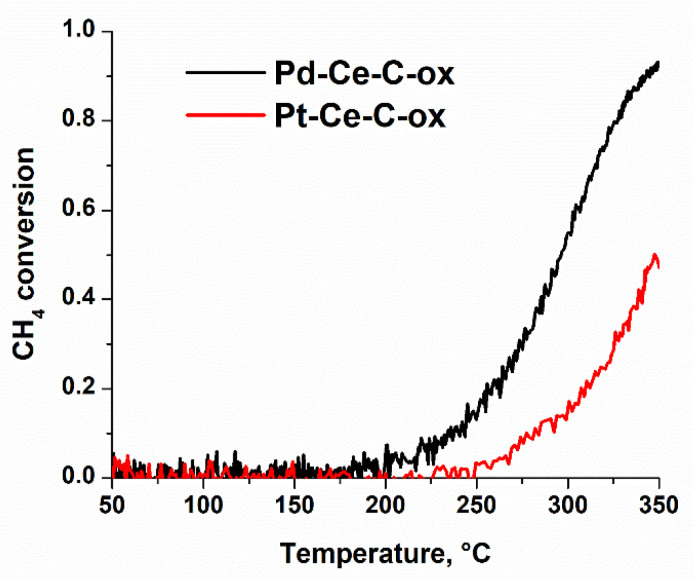
TPR-CH_4_+O_2_ data. The temperature dependences of CH_4_ conversion during the third heating run of the samples in the reaction mixture in the TPR-CH_4_+O_2_ mode for Pd-Ce-C-ox and Pt-Ce-C-ox catalysts.

**Table 1 materials-15-07485-t001:** The specific surface area (S_BET_) of the samples; and the structural parameters of the phases detected in the catalysts by XRD: D- average crystallite size, a(CeO_2_)—cell parameter, Δd/d—microstrain parameter, R_wp_—weighted profile R-factor.

Sample	S_BET_, m^2^/g	D(CeO_2_), nm	a(CeO_2_), Å	Δd/d(CeO_2_)	D(PdO), nm	D(Pt), nm	R_wp_, %
Pd-Ce-C	194	3(1) ^1^	5.432(1)	0.36(2)	8(1)	-	5.1
Pd-Ce-C-ox	213	8(1)	5.429(1)	0.16(3)	8(1)	-	4.6
Pt-Ce-C	168	3(1)	5.425(1)	0.30(3)	-	3(1)	4.3
Pt-Ce-C-ox	215	3(1)	5.434(1)	0.25(1)	-	6(3) ^2^	5.9
MWCNTs	180	-	-	-	-	-	-

^1^ Number in brackets shows a deviation in the last digit, e.g., 3(1) nm means 3 ± 1 nm. ^2^ Low intensity of the Pt peaks does not allow precise calculation of the size parameters.

## Data Availability

The data presented in this study are available on request from the corresponding author.

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
