# Peer review of "Pd-Ce-Ox/MWCNTs and Pt-Ce-Ox/MWCNTs Composite Materials: Morphology, Microstructure, and Catalytic Properties"

_materials, 2022, doi:10.3390/ma15217485_

Round 1
Reviewer 1 Report
Dear Authors
The manuscript is focused on the multi-walled carbon nanotubes (MWCNTs) were used for the preparation of Pd-Ce-Ox/MWCNTs and Pt-Ce-Ox/MWCNTs catalysts comprising the active components (Pd, Pt, ceria) as highly dispersed nanoparticles, clusters, and single atoms.
The following suggestion and comments should be taken:
1. The overall English needs to be improved. Please seek guidance from a native English speaker if possible ("the" "a", commas, plural form and others could be corrected).
2. The authors could insert more numerical data into the Abstract for enhancement of the manuscript.
3. Shouldn't substitutes for expensive Pt and Pd be used? Please explain.
4. The introduction section needs enhancement 1-3 sentences about carbon nanotubes and their modifications with heteroatoms. Please cite (1) Microsyst Technol 2021. https://doi.org/10.1007/s00542-021-05211-6 (2) Materials 2021, 14(9), 2448; https://doi.org/10.3390/ma14092448 (3) Renew. Sustain. Energy Rev.2017,68, 234–246
5. Could the authors include the standard deviation of the analysis?
6. I feel this paper has not given an extensive report on Brunauer-Emmett-Teller (BET) Surface Area Analysis and Barrett-Joyner-Halenda (BJH) Pore Size and Volume Analysis. What kind of pores the authors have please comment it.
7. Is SEM/EDS a good method to analyze elements such as C or O? Please explain.
8. Please add the discussion of the characterization by RAMAN spectroscopy.
9. Please add a table with the content of elements and 3-4 sentences to describe it.
10. EDX and XPS methods should have a different ratio of C:O in these two methods, what do the authors say about that?
11. Please add XPS spectra for C.
12. Could authors add AFM analysis of obtained materials?
13. Please answer the question in the comments: What do you think, are you have the material in or out of nanotubes. Do you have the encapsulation process in tubular structures?
14. Authors are suggested to describe some future plans in conclusions.
Reviewer 2 Report
I can recommend the publication of this manuscript after a minor revision.
Write keywords in alphabetical order.
Line 46: “.....and others [10–18].”
You will likely need to re-write your citation sentences, rather than simply replacing the numbers with Authors’ names. This is due to the fact that in order to give readers the maximum appreciation of how your work builds on previous results, each one of the cited sources should be discussed individually and explicitly to demonstrate their significance to your study. We ask that you use the authors' surnames as the subject of a verb, and then state in one or two sentences what they claim, what evidence they provide to support their claim, and how you evaluate their work. We also, therefore, ask that you avoid citing more than one reference in one sentence. This will give you a chance to discuss each reference separately.
What we are asking for is something like this: “Smith (2011) describes the development of a finite element model of hot forging and claims excellent agreement between the model and experiments. However, he tests only one operating condition, tunes his model by modifying the friction coefficient, and compares only the total tool force. A much more detailed comparison would be required to evaluate the precise conditions under which finite element modeling is truly accurate."
Insert more details about the statistical method applied and the corresponding software.
Explain with more details sentences from lines 192-194, 201-203, 232-234, 297-298, 315-317, 353-356.
Line 212: Fig. 2(a,b,e,f) – improve the resolution.
Line 267: give more details about these results.
Specify the limits of this study.
A minor mistake in Reference (Ref. [35]).
If possible, I recommend the following references: DOI: 10.1016/j.apcata.2021.118469; DOI: 10.1016/j.apsusc.2013.10.114.
Reviewer 3 Report
The authors have presented Pd/Pt-Ce-Ox/MWCNTs composite materials for morphological, microstructural and catalytic properties. Using CNT, MWCNT etc. based composites for different applications has been of interest in recent past. However, I would like authors a few issues before final decision;
1- Introduction should have literature review of some recent work related to CNT, MWCNT etc. composites.
2- Particle size has been calculated using XRD and crystallite size using TEM images. They appear to be same. Are they actually same. What method has been used for calculations.
3. Resolution of Figure 2 should be improved as in present form it does not show anything.
4- Last paragraph of the introduction is a mix of novelty, experimental methods and conclusions. Review it carefully.
Round 2
Reviewer 1 Report
The authors have addressed all comments and the manuscript can be published as is.